

# Patch and matrix characteristics determine the outcome of ecosystem engineering by mole rats in dry grasslands

Orsolya Valkó[1], András Kelemen[1,2], Orsolya Kiss[3] and Balázs Deák[1]

[1] Lendület Seed Ecology Research Group, Institute of Ecology and Botany, Centre for Ecological Research, Vácrátót, Hungary
[2] Department of Ecology, University of Szeged, Szeged, Hungary
[3] Institute of Animal Sciences and Wildlife Management, Faculty of Agriculture, University of Szeged, Hódmezővásárhely, Hungary

Corresponding author
Orsolya Valkó, valkoorsi@gmail.com

## ABSTRACT

**Background:** Burrowing mammals are important ecosystem engineers, especially in open ecosystems where they create patches that differ from the surrounding matrix in their structure or ecosystem functions.

**Methods:** We evaluated the fine-scale effects of a subterranean ecosystem engineer, the Lesser blind mole rat on the vegetation composition of sandy dry grasslands in Hungary. In this model system we tested whether the characteristics of the patch (mound size) and the matrix (total vegetation cover in the undisturbed grassland) influence the structural and functional contrasts between the mounds and the undisturbed grasslands. We sampled the vegetation of 80 mounds and 80 undisturbed grassland plots in four sites, where we recorded the total vegetation cover, and the occurrence and cover of each vascular plant species. We used two proxies to characterise the patches (mounds) and the matrix (undisturbed grassland): we measured the perimeter of the mounds and estimated the total vegetation cover of the undisturbed grasslands. First, we compared the vegetation characteristics of the mounds and the surrounding grasslands with general linear models. Second, we characterised the contrasts between the mounds and the undisturbed grassland by relative response indices (RRIs) of the vegetation characteristics studied in the first step.

**Results:** Species composition of the vegetation of the mounds and undisturbed grasslands was well separated in three out of the four study sites. Mounds were characterised by lower vegetation cover, lower cover of perennial graminoids, and higher diversity, and evenness compared to undisturbed grasslands. The contrast in vegetation cover between mounds and undisturbed grasslands increased with decreasing patch size. Increasing vegetation cover in the matrix grasslands increased the contrasts between the mounds and undisturbed grasslands in terms of total cover, perennial graminoid cover, diversity, and evenness. Our results suggest that mole rat mounds provide improved establishment conditions for subordinate species, because they are larger than other types of natural gaps and are characterised by less intense belowground competition. The ecosystem engineering effect, *i.e.*, the contrast between the patches and the matrix was the largest in the more closed grasslands.

## INTRODUCTION

Ecosystem engineer organisms create patches that differ from the surrounding matrix in their structure or ecosystem functions (*Jones, Lawton & Shachak, 1994*). In this way they alter the resource distribution in the landscape (*Mallen-Cooper, Nakagawa & Eldridge, 2019*; *Neilly, Cale & Eldridge, 2022*; *Valkó et al., 2022*). Previous syntheses on ecosystem engineers found that there are several factors determining the characteristics of the engineered patches, including the traits of the engineer, the habitat, climate and soil type (*Mallen-Cooper, Nakagawa & Eldridge, 2019*; *Root-Bernstein & Ebensperger, 2013*). Burrowing mammals are important ecosystem engineers, especially in open habitats (*Davidson, Detling & Brown, 2012*; *Reichman & Seabloom, 2002*; *Valkó et al., 2021*; *Whitford & Kay, 1999*). Many of these animals are endangered due to land use changes and various human activities, therefore it is crucial to understand the ecological functions they provide to support more effective protection (*Davidson, Detling & Brown, 2012*). Through their mound-building and burrowing activities, they move large amounts of soil and create sparsely vegetated patches that often have different vegetation compared to the surrounding habitat matrix (*Coggan, Hayward & Gibb, 2018*; *Mallen-Cooper, Nakagawa & Eldridge, 2019*).

Subterranean rodents inhabit open landscapes, including open woodland savanna in Africa (*e.g.*, African mole-rats or blesmols, Bathyergidae family, *Nevo, 1999*; *Visser, Bennett & van Vuuren, 2019*) and Mediterranean and continental steppes of Eurasia (*e.g.*, blind mole rats, Spalacidae family, *Németh et al., 2020*, *Nevo, 1999*; and zokors, Myotalpinae family *Zhang, Zhang & Liu, 2003*). They are highly specialised to a subterranean lifestyle and most of these species spend the majority of their time below ground. Therefore, their ecosystem engineering effect is different from burrowing mammals that feed and graze aboveground, like marmots (*Valkó et al., 2021*) or prairie dogs (*Winter, Cully & Pontius, 2002*). Subterranean rodents can alter vegetation composition by several mechanisms: (1) they generally feed on belowground plant organs, such as roots and bulbs which can decrease the abundance of certain plant species (*Šklíba et al., 2017*); (2) they create mounds with open soil surface that can play a role in vegetation dynamics as establishment gaps (*Reichman & Jarvis, 1989*); (3) their underground activity affects soil structure and several soil parameters, therefore vegetation as well (*Platt et al., 2016*; *Reichman & Seabloom, 2002*; *Zhang, Zhang & Liu, 2003*). Blind mole rats make food storages by hoarding bulbs, rhizomes and tubers and sometimes they do not eat all the stored plant organs so these can sprout later (*Szabó & Zimmermann, 2012*).

Subterranean rodents are ideal organisms for the study of ecosystem engineering, as they often create clearly visible mounds with sharp boundaries that are distinct patches in the grassland matrix (*Boldog, 2010*). African mole-rats are considered as ecosystem engineers in fynbos ecosystems, where their mounds are characterised by lower vegetation

cover and higher species richness compared to the surrounding matrix (*Davies & Jarvis, 1986*; *Hagenah & Bennett, 2013*; *Reichman & Jarvis, 1989*). The engineer effect of European blind mole rats was tested in one study conducted in temperate dry grasslands, where the vegetation of twelve mounds was compared with undisturbed grasslands (*Zimmermann et al., 2014*). No difference was found between the vegetation characteristics and species richness of the mounds and the surrounding grasslands, but some differences in the species composition were found. This weak engineering effect detected in European blind mole rats compared to African mole-rats might be due to the generally more accentuated effects of ecosystem engineers at lower latitudes (*Romero et al., 2015*). However, given the similarities in some ecological functions of African and Eurasian subterranean rodents, we believe that a more detailed analysis on the potential engineer effect of mole rats in temperate Eurasia is necessary.

The aim of our study was to evaluate the fine-scale effect of Lesser blind mole rat (*Nannospalax* (superspecies *leucodon*)) mounds on the vegetation composition of dry grasslands in Hungary. In this model system we tested whether the characteristics of the patch (mound size) and the matrix (total vegetation cover in the neighbouring undisturbed grassland) influence the structural and functional contrasts between the mounds and the undisturbed grasslands. We tested the following hypotheses: (i) Mounds have a more open vegetation structure, and different species composition and diversity patterns compared to the undisturbed grasslands. (ii) Vegetation of larger mounds is less affected by the edge effect so the structural and functional contrasts between the mounds and the undisturbed grasslands decrease with increasing mound size. (iii) The structural and functional contrasts between the vegetation of mounds and the undisturbed grasslands increase with increasing vegetation cover in the grassland matrix. As mole rats create a large number of mounds locally in an ecosystem type where gap dynamics are crucial driver of the vegetation composition, the study system is ideal for testing these hypotheses.

## MATERIALS AND METHODS

### Study system

The Lesser blind mole rat (*Nannospalax* (superspecies *leucodon*)) superspecies complex includes several morphologically very similar but genetically isolated species (*Csorba et al., 2015*; *Németh et al., 2020*). Besides genetic differences within the superspecies complex, involved taxa have the same ecological function and lifestyle, hence here we did not distinguish between them. These subterranean mammals are strictly protected and critically endangered in Central Europe (*Csorba et al., 2015*). They inhabit dry grasslands, old-fields, and sometimes also urban areas (*Németh, Moldován & Szél, 2020*). We sampled four study sites that all hold large populations of the mole rats. In Bagamér and Hajdúbagos sites the *Nannospalax* (*leucodon*) *transsylvanicus* taxon, while in Ásotthalom and Baja sites the *Nannospalax* (*leucodon*) *montanosyrmiensis* taxon occurs according to *Csorba et al. (2015)*. The study sites include many of the largest known populations of the species in Hungary and are characterised by pristine sandy grassland vegetation in a good conservation status (*Csorba et al., 2015*; *Németh, Moldován & Szél, 2020*). The study sites are located in the operation area of the Hortobágy National Park (Bagamér site—N

47.47028, E 21.95873, and Hajdúbagos site—N 47.41340, E 21.67606) and the Kiskunság National Park (Ásotthalom site—N 46.22235, E 19.67164, and Baja site—N 46.19643, E 18.99083). The Trans-Tisza Environmental, Nature Protection and Water Inspectorate approved this study (6646/08/2014). The characteristic vegetation of the study sites is dry sandy grassland, the dominant grass species are *Festuca pseudovina*, *F. rupicola*, *F. vaginata* and *Koeleria glauca*. The sites provide habitat to several protected grassland plant species, such as *Astragalus varius*, *Colchicum arenarium*, and the strictly protected *Pulsatilla flavescens* (*Borhidi, Kevey & Lendvai, 2012*).

## Sampling design

The study was performed in April 2020 and in each study site, we selected 20 mounds built by mole rats. We selected mounds that were built at least 1 year before the survey. We did not survey freshly built mounds with no vegetation and also, we did not consider mounds where the mound structure had been disintegrated. Instead of using a fix-sized sampling quadrat, we considered one mound as one sampling unit as in this case we could capture the potential within-mound variety in the vegetation. We adjusted a measuring tape along the mound edge and used the same measuring tape for delineating a control plot with the same size and shape as the mound. Control plots were designated in the undisturbed sandy grassland within a one meter distance from each mound. We recorded the perimeter of each mound.

We sampled the vegetation of 80 mounds and 80 undisturbed grassland plots: we recorded the total vegetation cover, and the occurrence and cover of each vascular plant species. Plant nomenclature follows the work of *Király (2009)*.

## Data analysis

We used two proxies to characterise the patches (mounds) and the matrix (undisturbed grassland): the perimeter of the mounds and the total vegetation cover of the undisturbed grasslands. We calculated the Shannon diversity and the evenness of the vegetation in each sampling unit.

First, we compared the vegetation characteristics of the mounds and the surrounding grasslands with general linear models, where the fixed factor was the microsite type (mound *vs* undisturbed grassland), and site was used as a random factor in the models. In the analysis, the following dependent variables were used: total vegetation cover, cover of perennial graminoids, species richness, Shannon diversity, and evenness.

Second, we characterised the contrasts between the mounds and the undisturbed grassland by relative response indices (RRIs, *Armas, Ordiales & Pugnaire, 2004*; *Perkins & Hatfield, 2014*) of the vegetation characteristics studied in the first step. RRIs were calculated based on the following equation: $RRI = (X_M - X_G)/(X_M + X_G)$; where $X_M$ is the value of a dependent variable (*e.g.*, Shannon diversity) in a mound and $X_G$ is the value of the same dependent variable in the undisturbed grassland plot paired with the adjacent mound plot. Value of RRI ranges between −1 and +1. The closer |RRI| is to 1, the higher the contrast between the mounds and the undisturbed grasslands. With generalized linear mixed models (GLMMs), we tested the effects of patch size (fixed factor) and total

vegetation cover in matrix grassland (fixed factor) on the contrasts between the mounds and the undisturbed grasslands (*i.e.*, |RRI|s of the studied vegetation characteristics). Study site was used as random factor. GLMs and GLMMs were calculated using SPSS 17.0.

We applied non-metric multidimensional scaling (NMDS) using Bray-Curtis index of dissimilarity to test differences in the species composition of the mounds and undisturbed grasslands in CANOCO 5.0 program (*Ter Braak & Šmilauer, 2012*). We made the calculations for each site separately.

## RESULTS

We recorded in total 112 vascular plant species, out of which 102 species occurred on mounds and 93 in undisturbed grassland plots. Nineteen species occurred exclusively on mounds and ten exclusively in the grassland plots. Sixty-four species were more frequent on mounds than in the undisturbed grasslands, 18 occurred with the same frequency, and 30 species were more frequent in the undisturbed grasslands. Out of the six protected plant species recorded at the study sites, one occurred only on mounds (*Pulsatilla flavescens*), four occurred both on mounds and grassland plots (*Colchicum arenarium, Dianthus serotinus, Onosma arenaria, Stipa borystenica*) and one only in undisturbed grassland plots (*Astragalus varius*). For the complete list of the recorded species, please see Appendix 1.

Species composition of the vegetation of the mounds and undisturbed grasslands was well separated in three of the study sites (Hajdúbagos, Bagamér and Ásotthalom), whilst it showed considerable similarity in the case of the Baja site (Fig. 1). The dominant grass species (*Festuca* spp.) and some typical perennial forb species (such as *Thymus glabrescens, Potentilla arenaria, Euphorbia cyparissias*) of the studied dry grasslands characterised the undisturbed grasslands. Vegetation of mounds were characterised by several disturbance-tolerant species (such as *Erophila verna, Eryngium campestre, Poa bulbosa, Rumex acetosella, Vicia lathyroides*).

Mounds were characterised by lower vegetation cover (F = 87.168, $p = 0.003$), lower cover of perennial graminoids (F = 93.503, $p = 0.002$), higher Shannon diversity (F = 16.422, $p = 0.027$) and evenness (F = 15.780, $p = 0.029$) compared to undisturbed grasslands (Fig. 2). Species richness on the mounds and in the undisturbed grasslands was not different (F = 6.820, $p = 0.080$).

The average perimeter of the mounds was 2.18 m ± 0.88 SD, and the total vegetation cover in the undisturbed grasslands was 66.94% ± 14.17 SD. There was a high contrast in total vegetation cover between small patches and the matrix, which decreased with increasing patch size (Table 1). RRI calculated for total vegetation cover was the lowest in small patches and increased with increasing patch size. The RRIs calculated for perennial graminoid cover, species richness, Shannon diversity and evenness were not affected by patch size (Table 1, Appendix 2).

Increasing vegetation cover in the matrix grasslands increased the contrasts between the vegetation of the mounds and undisturbed grasslands in terms of total cover, perennial graminoid cover, Shannon diversity and evenness (Table 1, Appendix 3). RRIs calculated for total vegetation cover and perennial graminoid cover were lower in more closed grasslands, indicating larger contrasts between the mounds and the undisturbed grasslands

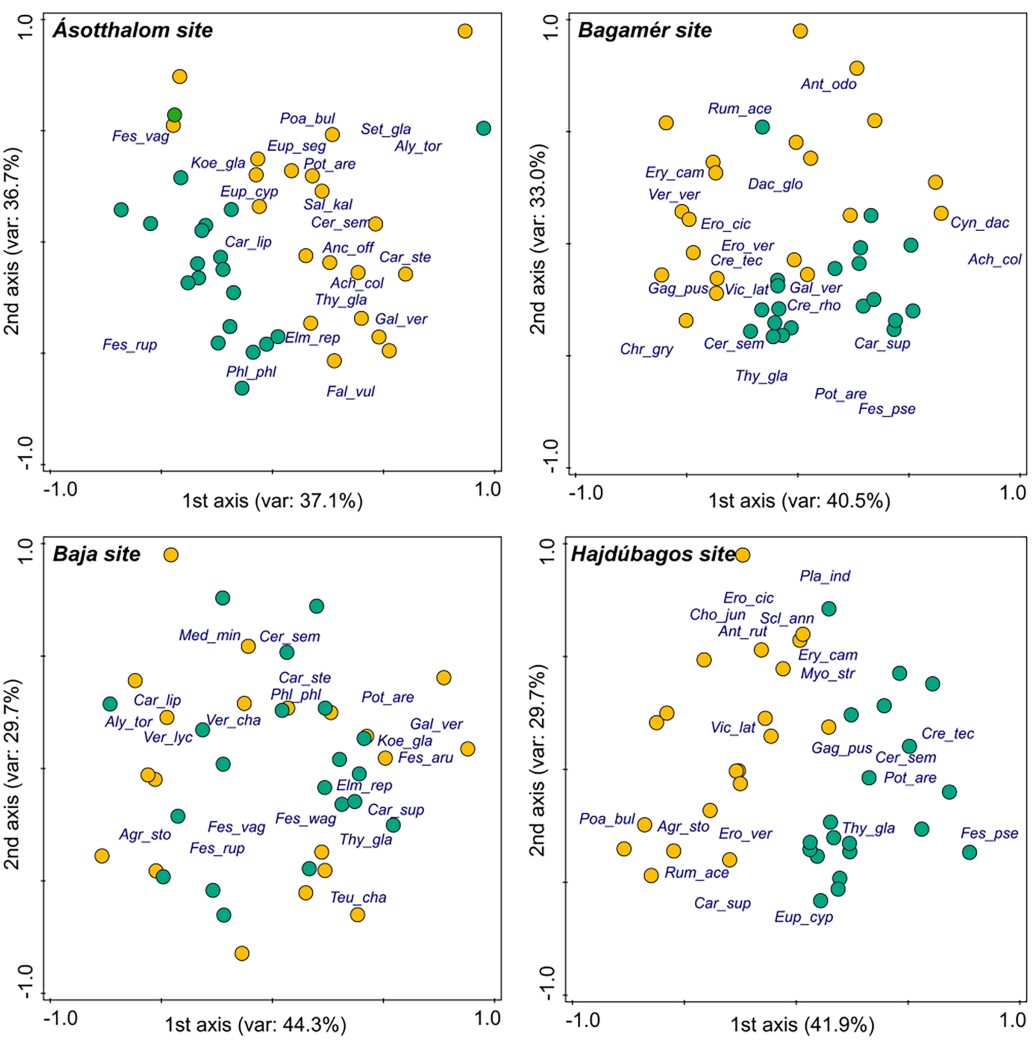

**Figure 1** **Differences in the species composition of mounds and undisturbed grasslands in the four studied sites (NMDS ordination).** We plotted the 20 most abundant species at each site on the panels. Species names are abbreviated using the first three letters of the genus and species names. Yellow circles denote mole rat mounds, green circles denote undisturbed grasslands.

than in more open grasslands. RRIs calculated for Shannon diversity and evenness increased with increasing cover of the matrix grasslands, indicating that mole rat mounds are more diverse than the undisturbed grasslands in the more closed grasslands (Appendix 3).

# DISCUSSION

We found that Lesser blind mole rats created patches with different structural attributes (lower vegetation cover, lower cover of perennial graminoids) compared to the surrounding sand grassland. The vegetation of the mounds and undisturbed grasslands was different in terms of total vegetation cover, cover of perennial graminoids as well as Shannon diversity and evenness, which confirmed our first hypothesis. This suggests that the mole rats play an important role in the gap dynamics in the study system. We found a
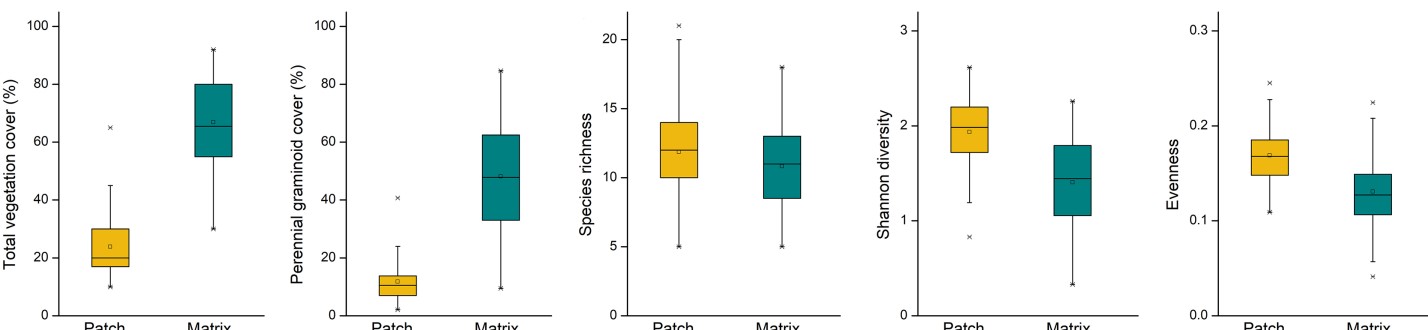

**Figure 2 Vegetation characteristics on the patches (mole rat mounds) and matrix (undisturbed grasslands).** Yellow boxes denote mole rat mounds, green boxes denote undisturbed grasslands.

**Table 1 The effect of patch size and matrix cover (fixed factors) on the contrasts between the vegetation of the mounds and the grassland matrix.** Significant effects are marked with boldface.

| | Patch size | | | Matrix cover | | |
|---|---|---|---|---|---|---|
| | **Direction** | **F** | **p** | **Direction** | **F** | **p** |
| Total cover, RRI | ↑ | **6.355** | **0.014** | ↓ | **31.240** | **0.000** |
| Species richness, RRI | | 0.715 | 0.401 | | 0.294 | 0.589 |
| Perennial graminoid cover, RRI | | 0.360 | 0.550 | ↓ | **9.527** | **0.003** |
| Shannon diversity, RRI | | 0.219 | 0.641 | ↑ | **4.135** | **0.045** |
| Evenness, RRI | | 0.012 | 0.912 | ↑ | **9.303** | **0.003** |

Note:
Contrasts were expressed by the relative response indices (RRIs) calculated between the vegetation characteristics of the mounds and the undisturbed grasslands.

more open vegetation with lower cover of perennial graminoids on the mounds, which is a general pattern observed on mounds of various burrowing mammals (*e.g.*, root-rats: *Asefa et al., 2022*; prairie dogs: *Winter, Cully & Pontius, 2002*; pikas: *Wesche, Nadrowski & Retzer, 2007*; marmots: *Valkó et al., 2021*). These are notable effects considering the small size of the mounds. Note that the vegetation of the control plots might also have been influenced by the underground activity of the mole rats (*e.g.*, root consumption), which is a potential limitation of our study. However, this potential bias is consistent in all the 80 pairs of mounds and control plots, as these paired treatments were always situated at the same distance (1 m) from each other.

In the studied sandy grassland ecosystems, the vegetation is open, but there is an intense belowground competition between the perennial graminoids and other subordinate plant species (*Borhidi, Kevey & Lendvai, 2012*). It is possible that the gaps created by mole rats can provide improved establishment conditions for subordinate species, because the mounds are larger than other types of natural gaps. Also, as mole rats feed on roots and other belowground organs (*Corbet, 1984*), they can locally decrease the belowground competition which gives an establishment advantage to subordinate species over perennial graminoids. Similar results were obtained in a study about the effect of plateau zokor (*Myospalax fontanierii*) mounds on the vegetation of alpine meadows in the Tibetean Plateau (*Zhang, Zhang & Liu, 2003*). A study on the root-rat (*Tachyoryctes macrocephalus*)

mounds in Ethiopian grasslands found that the mounds were characterised by lower cover of the dominant species and higher cover of subordinate plants (*Šklíba et al., 2017*), which is also in line with our findings. The differences between the vegetation of the mole rat mounds and the undisturbed grasslands is interesting also because of the subterranean lifestyle of the mole rats. In most burrowing mammals, mounds are not only biogeomorphological features but the mammals' activities on the mound surface also shape vegetation composition, *e.g.*, by trampling and manuring (*e.g.*, foxes, *Godó et al., 2018*; marmots, *Valkó et al., 2021*; pikas, *Yoshihara et al., 2010*), but this is not the case in mole rats that spend most of their life underground.

Compared to the only other published study on the ecosystem engineer effect of mole rat species in temperate Eurasia (*Zimmermann et al., 2014*), we found more and stronger evidence for the engineer effects. The detected weaker evidence of the engineer effect in the previous study can be either a result of the particular study design (one study site, small sample size, fixed plot size in the other study) or by the slightly different habitat types considered (sandy grasslands in our study and loess steppes in *Zimmermann et al., 2014*).

The species composition of the mounds and undisturbed grasslands was not different when considering all the sites together; however, looking at the site level we found marked differences in three of the four study sites. This finding supports our first hypothesis and also suggests that the effect of mole rats on the vegetation should be considered at the local and not the regional scale. This is in line with another study on the effect of fine-scale environmental heterogeneity on the species composition of grasslands, where the effects of environmental heterogeneity were more pronounced on the local than on the regional scale (*Deák et al., 2021*).

Most of the species recorded in the study (74%) occurred both on the mounds and in the intact grasslands. This implies that mounds are not unique establishment microsites for the majority of plant species; however, mounds can provide improved establishment opportunities for subordinate species, due to the low level of competition by perennial graminoids. There was no difference between the species richness of the mounds and the undisturbed grasslands, but vegetation patches on the mounds were more diverse and the species were more evenly distributed compared to the undisturbed grassland matrix.

Our second hypothesis, *i.e.*, that the contrasts between the mounds and the undisturbed grasslands decrease with increasing mound size, was partly supported: in case of total vegetation cover, smaller mounds were more different from the undisturbed grasslands than larger ones. This suggests that the height of the mounds (approximately 20 cm) and their steep slopes provide a sharp vegetation boundary which prevents the clonal growth of the surrounding vegetation on the mound. The colonization of the mounds by plants is probably driven by random dispersal processes (*i.e.*, seed rain), and a higher number of incoming diaspores can be expected on the larger surface of larger patches. Also, besides total vegetation cover, we found that mound size did not affect the other studied variables. Even there was variation in mound size they were rather small so the effect of patch size might be relevant in other scales.

We confirmed our third hypothesis as the contrasts between the vegetation of mounds and undisturbed grasslands were higher in the more closed grasslands. This suggests that

the importance of the ecosystem engineering effect is the highest in the more closed grasslands, where the engineer organisms increase more the structural and functional heterogeneity of the ecosystem. For the conservation of the plant species associated to dry grasslands, creating proper establishment microsites is crucial (*Klaus et al., 2018*). Our results suggest that mounds of mole rats can provide suitable establishment microsites for approximately 91% of the species pool of the studied sandy dry grasslands. Thus, they might be potentially feasible as establishment gaps in restoration projects or when introducing particular rare species (*Kiss et al., 2021*; *Limb et al., 2010*). The spatio-temporal dynamics of the creation and the re-vegetation of the mounds can be an important driver of establishment of subordinate species in the studied grassland ecosystems. These results highlight that the protection of these endangered subterranean rodents is crucial also for maintaining the vegetation dynamics and ecosystem functioning of their habitats. Further studies are needed for testing the effectiveness of natural gaps created by Lesser blind mole rats in increasing species richness during restoration.

## CONCLUSIONS

We found that the subterranean mole rats create patches in temperate sandy grasslands that differ from the undisturbed grasslands in species composition and vegetation characteristics. The contrast between the vegetation of the mounds and undisturbed grasslands were the sharpest in grasslands with more closed vegetation cover. We suggest that the contrasts between the patches and the matrix, which was proposed in this study as a proxy for the strength of the engineering effect, can be a useful variable also in other studies. The effects of patch and matrix characteristics on the contrasts should be studied in other ecosystems and other organisms for a deeper understanding of the mechanisms beyond ecosystem engineering.

## ACKNOWLEDGEMENTS

We are grateful to László Szél and Szabolcs Lengyel for interesting discussions about the study topic.

### Funding

The authors were funded by the Hungarian Research, Development, and Innovation Office (Grant Numbers: FK 124404 (Orsolya Valkó), KKP 144096 (Orsolya Valkó), and FK 135329 (Balázs Deák)). András Kelemen was funded by the Bolyai János Scholarship of the Hungarian Academy of Sciences. The funders had no role in study design, data collection and analysis, decision to publish, or preparation of the manuscript.

### Grant Disclosures

The following grant information was disclosed by the authors:
Hungarian Research, Development, and Innovation Office: FK 124404, KKP 144096, FK

135329.
Bolyai János Scholarship of the Hungarian Academy of Sciences.

## Competing Interests

The authors declare that they have no competing interests.

## Author Contributions

- Orsolya Valkó conceived and designed the experiments, performed the experiments, analyzed the data, prepared figures and/or tables, authored or reviewed drafts of the article, and approved the final draft.
- András Kelemen conceived and designed the experiments, performed the experiments, authored or reviewed drafts of the article, and approved the final draft.
- Orsolya Kiss conceived and designed the experiments, performed the experiments, authored or reviewed drafts of the article, and approved the final draft.
- Balázs Deák conceived and designed the experiments, performed the experiments, analyzed the data, prepared figures and/or tables, authored or reviewed drafts of the article, and approved the final draft.

## Field Study Permissions

The following information was supplied relating to field study approvals (*i.e.*, approving body and any reference numbers):

The Trans-Tisza Environmental, Nature Protection and Water Inspectorate approved this study (6646/08/2014).

## Data Availability

The raw data are available in the Supplemental Files.

## Supplemental Information

Supplemental information for this article can be found online at http://dx.doi.org/10.7717/peerj.14582#supplemental-information.

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
