# Peer review of "Patch and matrix characteristics determine the outcome of ecosystem engineering by mole rats in dry grasslands"

_PeerJ, doi:10.7717/peerj.14582_

## Round 0.1 · original submission · Major Revisions

Dear Authors
Three reviewers have analyzed your manuscript and expressed a highly positive opinion and I agree with them. However, they highlighted several points in which the manuscript requires revision or clarification (see in particular reviewer 3).

Please also pay attention to the files attached to the review.

I'm looking forward to receiving the revised version of this manuscript.

Reviewer 1 ·

Basic reporting

Some important literature references and the information contained therein are not or incompletely presented in the article. In some cases, technical terms or the name of the examined group of animals were not used accurately or correctly.

The exact presentation of the contested parts can be found below:
The paragraphs between lines 62 and 86 needs rewording for the reasons detailed below;

Although in common English the members of both Bathyergidae and Spalacinae are called mole rats, there is a huge distance between the two groups both evolutionarily and taxonomically, so I would not recommend summarizing them under this name. Since we are talking about two very different groups of rodents, it is extremely confusing to lump them together under the same name to give a discussion about both. In my opinion, the term ‘subterranean rodent’ would be much more appropriate, or I would suggest using their scientific family/subfamily name.
(The common English name of Bathyergidae is African mole rat or blesmols.)

I understand and see the concept in the introduction comparing the ecological effects of African and Eurasian ground-dwelling rodents on vegetation. However, the difference in evolutionary systematics between the two rodent groups is so great that I would definitely recommend rethinking and reworking the logic in the introduction in a way that goes beyond the similarities caused by parallel evolution and takes into account the evolutionary point of view.

Prairie dogs and marmots are burrowing rodents, while African mole rats and blind mole rats are subterranean rodents. Please check the very special literature of subterranean rodents regarding these questions (e.g. Nevo 1999 – Mosaic Evolution of Subterranean Mammals). Although we find an almost continuous transition between the two lifestyles in the way of life of the different rodents, the two lifestyle differ significantly in terms of adaptations, behavior and ecology. The relevant literature and research clearly separates and discuss apart the two lifestyles. Therefore, comparing the two lifestyles in this way is quite unusual for researchers who are studying these rodents.

Mole rats (whatever it means, please specify what you mean exactly? I suggest using subterranean rodents or blind mole rats if you are just talking about European representatives) can affect vegetation in more ways than those listed here. 1) Theirs underground activity affects soil structure and therefore vegetation as well. 2) Blind mole rats makes food storages hoarding bulbs, rhizomes and tubers and sometimes they do not eat it all, they just leave it there. These vegetative plant parts can sprout later. Thereby blind mole rats contributing to the spread of certain plants. 3) and 4) are those points mentioned in the manuscript.

In line 73, again it is very confusing that it is not clear if the authors are talking about members of Bathyergidae and Spalacinae together or just African mole rats (Bathyergidae). Please, use African mole rat for Bathyergidae and blind mole rats or Eurasian blind mole rat for Spalacinae. Please, not lump them together.

In Line 88, the proper name for the superspecies containing the European members of the genus Nannospalax in Lesser blind mole rat, Nannospalax (superspecies leucodon)

Experimental design

There are minor inaccuracies and uncertainties in the materials and methods chapter, which need clarification and clarification.

According to Csorba et al. 2015 (I suggest to check this publication, as it is the best summary of the status of Central European blind mole rats), the largest blind mole rat population in Hungary can be found near the settlement of Hajduhadhaz. However, despite the manuscript claims that the largest known localities were investigated, it is not included in the list of the investigated habitat. Please clarify this.

I am not very sure, but if I know it right, no blind mole rat populations located in the territory of the Hortobagy National Park or in the territory of the Kiskunsag National Park, but in the operation area of the two mentioned national park directorate. Please clarify this.

Based on what criteria were it possible to determine that a mound is one year old? As far as I know, there is no exact method for determining the age of blind mole rat mounds. Furthermore, the features of the habitats (e.g. pedological features) and previous weather events greatly influence the morphological changes and degradation of the mounds.

Validity of the findings

no comment

Additional comments

The manuscript presents a very interesting and important research, which examines the role of Eurasian blind mole rats as ecosystem engineers, which has been assumed for a long time but has hardly been investigated until recently. It is the most thorough and comprehensive investigation in this field so far, so its publication may contribute to the expansion of our knowledge. Since, the ecological role of blind mole rats in theirs ecosystem is an area with lack of knowledge.

The results are not only important in terms of filling the gaps in our knowledge in the hitherto little-known and data-deficient Central European blind mole rats, but since they are seriously endangered species, it can also contribute to theirs conservation.

·

Basic reporting

No comment

Experimental design

Could the authors perhaps expand a bit on how they selected the mole rat mounds that they measured? It would be helpful to mention the age of the mounds - for example fresh mounds would not have any vegetation, and this will change with time, thus I think it is quite important to explain which criteria was used to select the mounds.

I am also wondering about the circumference of the mounds mentioned in the results. A circumference of more than 2 meters seems a bit excessive, or are the mounds flattened out? In the discussion you say that they are 10cm high and have steep sides, which sounds quite small.

Validity of the findings

The findings is congruent with other studies of disturbed environments, thus valid.

Additional comments

I think this is an interesting article, it improved and refined field techniques previously used to assess the vegetation structure on mole rat mounds, and differences were found in localised vegetation between mounds and the surrounding areas. It is very important to report positive effects that rodents can have on plant communities (or in any other aspect), especially ones that usually have a reputation as being destructive.

Reviewer 3 ·

Basic reporting

This is the review of the manuscript entitled "Patch and matrix characteristics determine the outcome of ecosystem engineering: a case study on the vegetation of mole rat mounds" by Valkó and colleagues submitted to PeerJ. This is a very interesting topic related to the burrowing behaviour of subterranean mammals with a particular interest in rodents. Importantly, this issue has not been studied adequately in European blind mole rats yet. Considering this, I think that this topic is worth sharing with other researchers and suits the selected journal well. Nevertheless, I think that the manuscript could, and should be, improved in several important aspects to better present the obtained data. I have provided ideas on what and how could be improved in my review.

The manuscript is well written and conforms to the professional standards of the selected journal. The introduction provides important background to understand the topic. Nevertheless, I would encourage the authors to add some very relevant references to go deeper into the topics. Importantly, there are several important papers that are not mentioned in their text, even though they relate directly to the given topic. These papers should be definitely cited in the proposed manuscript; I truly believe they will greatly improve its quality! For instance, the work of Zhang et al. (2003) shows the ecosystem role of zokors of the genus Myospalax, the species closely related to the focal species of their study. Besides, there is an important study on pocket gophers, indicating their important ecological role in North America (see Reichman and Seabloom 2002). Similarly, work on the giant root-rat from Ethiopian Highlands demonstrates pretty well its crucial role in the Afroalpine ecosystems (Šklíba et al. 2016, Asefa et al. 2022).
1. Asefa, A., Reuber, V., Miehe, G., Wondafrash, M., Wraase, L., Wube, T., ... & Schabo, D. G. (2022). The activity of a subterranean small mammal alters Afroalpine vegetation patterns and is positively affected by livestock grazing. Basic and Applied Ecology.
2. Reichman, O. J., & Seabloom, E. W. (2002). The role of pocket gophers as subterranean ecosystem engineers. Trends in Ecology & Evolution, 17(1), 44-49.
3. Šklíba, J., Vlasata, T., Lövy, M., Hrouzkova, E., Meheretu, Y., SILLERO‐ZUBIRI, C., & Šumbera, R. (2017). Ecological role of the giant root‐rat (Tachyoryctes macrocephalus) in the Afroalpine ecosystem. Integrative Zoology, 12(4), 333-344.
4. Zhang, Y., Zhang, Z., & Liu, J. (2003). Burrowing rodents as ecosystem engineers: the ecology and management of plateau zokors Myospalax fontanierii in alpine meadow ecosystems on the Tibetan Plateau. Mammal Review, 33(3‐4), 284-294.

Moreover, I would like to stress one important point. The authors mention repeatedly that mole-rats inhabit arid and semi-arid regions in Africa. This is definitely not true. These rodents, belonging to the families Bathyergidae and Spalacidae, live primarily in seasonal habitats. In the case of African mole-rats, they are mostly found in open woodland savannahs. Blind mole rats occupy primarily Mediterranean steppe environments. These habitats might be dry, especially during relatively long dry seasons, but they are not definitely classified as arid regions. Of course, there are some exceptions to this general pattern but it cannot be written that these rodents inhabit arid and semi-arid regions in general!

Please pay attention and name correctly African mole-rats (with a hyphen) and blind mole rats (without a hyphen).

The manuscript is well structured and two figures and one table follow the text well. Nevertheless, I have some suggestions on how to improve the figures and table to improve the clarity of the text (see my comments in other sections).

Experimental design

Although the research questions are quite well defined, I think this might be improved. Whereas hypotheses I and II are clearly understandable, I am wondering what hypothesis III means exactly. What do the authors mean by the term “increasing canopy” of the grassland habitats and how is that relevant/should affect functional contrasts between the mounds and the undisturbed grasslands?

Regarding the Methods, I have several important questions that need to be clarified. I understand that the authors had selected 20 mounds. How did you know that the mounds are exactly one year old? This is a crucial aspect of the whole study, as the vegetation can change substantially between the years. Therefore, all the selected mounds must be of the same age. I think that you should provide more details about how you selected them. Is 1m far enough to consider it undisturbed by mole rats? Mole rat burrow systems are rather dynamic in time and space and it might be more suitable to use more distant sites as control sites.

Regarding data analysis, I have the following questions/suggestions. In GLMMs, one of the predictors was the patch (=mole rat mound) size. Was it a numerical variable indicating the size of each mound in cm? If so, I think it would be great to add a new figure showing the relationship between RRI and mound size. It will greatly improve the presentation of your result. I think that the authors should describe more in detail why they have chosen non-metric multidimensional scaling to present differences in the species composition between the mounds and control grassland sites.

Validity of the findings

In general, the results are well-written and clearly understandable. I am wondering whether the authors have tried to test all 4 focal sites together in one multidimensional space. In the current version of the manuscript, the 4 sites are shown separately in Fig. 1. It is not stated what light and dark grey circles denote. Apparently, this information is missing from the figure legend. Why do not some of the circles have any labels? In figure 2, I would remove labels A, b, C, and D - you do not refer to them across the manuscript.

Based on the average perimeter of the mounds, it seems that mounds were relatively big (ca. with a radius of 35cm). Is that the normal size of the mound? I am not familiar much with Hungarian blind mole rats, but some other Nannospalax species from the Middle East usually have smaller mounds.

I think that you should elaborate on the last paragraph of the results. The results of the GLMMs are not described properly. You should put more effort to describe the effect of the predictors and how they affect RRIs. It is rather difficult for me to follow your results of the GLMMs, even though they represent the core of your work.

I think that the Discussion needs to be elaborated to interpret your findings in a clearer way. For instance, please specify what exactly your hypothesis II is. It will be easier to follow your findings and interpretation of them. As I mentioned in my review, it is hard to understand the hypothesis I if you mention canopy in open grassland habitats.

Additional comments

Please find below my specific comments:
L21-22: I do not agree with you that burrowing mammals are important engineers, especially in the arid and semi-arid regions. I think several studies have shown the important ecological function of subterranean mammals in both non-arid climates.
L42: should be a new paragraph Discussion/Conclusion
L56: the same comment as at the beginning.
L62: Mole-rats denote African mole-rats (Bathyergidae), "mole rats" should refer to blind mole rats. Please be consistent in this throughout the text. They do not inhabit arid and semi-arid regions in Africa and Eurasia! The majority of the distribution lies outside the arid (semi-arid) regions!
L83: Reference Romero et al. 2015 is not on the list.
L88: provide the proper scientific name for your focal species.
L93: What do you mean by "different vegetation"? Should be elaborated.
L97: increase.
L109: Are those four localities inhabited by the same blind mole rat species?
L117: The TransTisza?
L121: Astragalus varius
L126: Was each of these 20 mounds built by a different individual mole rat?
L209: You should refer to work on more closely related species of subterranean mammals - African mole-rats, other splacids, pocket gophers etc. These are more relevant in terms of their ecological functions than pikas or prairie dogs which are mostly fossorial species. Blind more tas and African mole-rats are highly subterranean species!
L233: Were the differences in the species composition of the mounds and undisturbed grasslands calculated for all 4 sites together?
L249: Please specify your hypothesis II here. It will be easier to follow your results and interpretation of them.
L251: After one year, the mounds are much lower than when they are made by a mole rat. This might highly impact the succession rate of plants growing on/in the mounds. Considering the relatively large size of your mounds, I am not sure if the slope is steep enough to provide a sharp vegetation boundary. Can you please clarify this?
L259: I think that the first sentence is very hard to follow. I would suggest rephrasing it to make it clearer.
L278: What do you mean by "closed canopy cover"? Need to be clarify.

---

## Round 0.2 · Minor Revisions

Dear Authors,

Both reviewers are satisfied with the changes you have made to the manuscript; there are only small things left to fix in the text.

Reviewer 1 ·

Basic reporting

Thanks to the revision, the manuscript has been greatly improved, so it meets the expectations set by the journal.
I only have a few small comments that need to be corrected;
- The first is that the correct English name of the Lesser blind mole rat has not been corrected everywhere in the text, e.g. line 121.
- The genetically distinct blind mole rat taxa were not correctly named for each study site. The sites of occurrence of N. (l.) transsylvanicus are accurate, but in the vicinity of Baja and Ásotthalom, N. (l.) montanosyrmiensis occurs actually (see Csorba et al. 2015).

Experimental design

I found the revised manuscript to be adequate in this regard. I have no additional comment.

Validity of the findings

I found the revised manuscript to be adequate in this regard. I have no additional comment.

Additional comments

The manuscript has been greatly improved thanks to thorough revision, so I consider it acceptable for publication after correcting the two minor errors indicated above.

·

Basic reporting

No comment

Experimental design

Good

Validity of the findings

Good

Additional comments

The authors have addressed all my previous concerns, I noted some minor grammatical errors

Line 62: remove 'their' between ...to support.. and ... more effective protection...
Line 151: change 'in one meter' to 'within a one meter'
Line 165: insert 'the' before 'microsite type'
Line 166: insert 'a' between 'used as' and 'random factor'

Line 121: western or Lesser blind mole-rat? You changed western to lesser in line 102
Line 235: western or lesser?

---

## Round 0.3 · accepted · Accept

Dear Authors,

Thank you for the revised version of your paper. The manuscript is now ready to be published